An atlas of rational genetic engineering strategies for improved xylose metabolism in Saccharomyces cerevisiae

Vargas Beatriz de Oliveira
dos Santos Jade Ribeiro
Pereira Gonçalo Amarante Guimarães goncalo@unicamp.br
de Mello Fellipe da Silveira Bezerra
Department of Genetics, Evolution, Microbiology, and Immunology, Institute of Biology, Universidade Estadual de Campinas , Campinas , Brazil
Carter Dee
Electronic publication date: 2023 Nov 28
Publication date: 2023
Volume: 11
Electronic Location ID: e16340
Received 2023 May 18; Accepted 2023 Oct 3
Copyright: ©2023 Vargas et al.
Copyright year: 2023
Copyright holder: Vargas et al.
License: This is an open access article distributed under the terms of the Creative Commons Attribution License, which permits unrestricted use, distribution, reproduction and adaptation in any medium and for any purpose provided that it is properly attributed. For attribution, the original author(s), title, publication source (PeerJ) and either DOI or URL of the article must be cited.
License URL: https://creativecommons.org/licenses/by/4.0/

Keywords: Xylose, Saccharomyces cerevisiae, Metabolic engineering, Cellulosic ethanol, Yeast, PPP, CRISPR-Cas9, Bioenergy, Bioethanol, Gene editing

Funding: The National Agency of Petroleum, Natural Gas and Biofuels (ANP), Brazil The P,D&I Clauses; the Sinochem Petróleo Brasil Ltda; and Conselho Nacional de Desenvolvimento Científico e Tecnológico (CNPq) 142340/2020-0 This study was financed by the National Agency of Petroleum, Natural Gas and Biofuels (ANP), Brazil, associated with the investment of resources from the P,D&I Clauses; the Sinochem Petróleo Brasil Ltda; and Conselho Nacional de Desenvolvimento Científico e Tecnológico (CNPq) through a scholarship to Jade R dos Santos (142340/2020-0). The funders had no role in study design, data collection and analysis, decision to publish, or preparation of the manuscript.

==============================
Xylose is the second most abundant carbohydrate in nature, mostly present in lignocellulosic material, and representing an appealing feedstock for molecule manufacturing through biotechnological routes. However, Saccharomyces cerevisiae—a microbial cell widely used industrially for ethanol production—is unable to assimilate this sugar. Hence, in a world with raising environmental awareness, the efficient fermentation of pentoses is a crucial bottleneck to producing biofuels from renewable biomass resources. In this context, advances in the genetic mapping of S. cerevisiae have contributed to noteworthy progress in the understanding of xylose metabolism in yeast, as well as the identification of gene targets that enable the development of tailored strains for cellulosic ethanol production. Accordingly, this review focuses on the main strategies employed to understand the network of genes that are directly or indirectly related to this phenotype, and their respective contributions to xylose consumption in S. cerevisiae, especially for ethanol production. Altogether, the information in this work summarizes the most recent and relevant results from scientific investigations that endowed S. cerevisiae with an outstanding capability for commercial ethanol production from xylose.

Introduction

Modern globalization has been historically structured on the use of energy provided by fossil sources; however, due to the high industrialization rates and a continual increase in world energy demand, a climate emergency and fuel crisis seem to be the main issues that humanity will face in the future if alternative and renewable energy sources are not fully explored. Within this context, biorefineries—which use lignocellulosic biomass feedstock to produce a variety of molecules, such as ethanol—are important vectors for the generation of sustainable biofuels, envisioning the total or partial replacement of fossil-based fuels. Ethanol is the most used biofuel in the world; in 2021, 103.4 million liters were commercialized worldwide (Renewable Fuels Association, 2022), and, due to the growing concern for environmental preservation in recent years, more investments are being made in order to develop new technologies that economically warrant the renewable energy industry.

For first generation (1G) ethanol production, the yeast Saccharomyces cerevisiae is used for the fermentation of hexoses, such as glucose, available from the main product of designated crops. From this process, lignocellulosic residues are generated, comprising a material with neglected sugar content that can be further exploited to produce second-generation (2G) ethanol (Santos et al., 2015). The concentration of such carbohydrates varies depending on the crop used; however, one of the main monomers found in the hemicellulose fraction is xylose (Chandel et al., 2021). The use of a microorganism with the ability to consume both hexoses (glucose) and pentoses (xylose) would be ideal to explore all the energy potential from such biomass. However, natural xylose-fermenting yeasts, such as Scheffersomyces stipitis and Spathaspora passalidarum, do not have the same fermentative capacity, tolerance to high levels of ethanol, or robustness shown by S. cerevisiae (Balat, 2011; Cadete et al., 2016). Thus, one of the main challenges for the efficient production of 2G ethanol is the insertion of xylose assimilation pathways into S. cerevisiae, since it does not consume this pentose naturally (Wang & Schneider, 1980).

There are two known pathways for xylose metabolism, each from distinct evolutionary origins and harboring different biochemical properties, that can be used for heterologous expression in S. cerevisiae: the oxidoreductive (XR-XDH) (Ho, Chen & Brainard, 1998) and the xylose isomerase (XI) (Brat, Boles & Wiedemann, 2009) pathways (Fig. 1). However, the insertion of these pathways alone does not guarantee an optimal xylose fermentation, as several works have already stated (Sarthy et al., 1987; Amore, Wilhelm & Hollenberg, 1989; Moes, Pretorius & Zyl, 1996; Gárdonyi & Hahn-Hägerdal, 2003). In this context, efforts have been made to endow superior xylose-fermenting ability in S. cerevisiae, aiming at the efficient expression of the genetic architecture related to this phenotype. Different genetic mapping strategies allow the understanding of the gene network underlying such traits, and genetic engineering enables the development of yeast strains that can be used in the industry by increasing the productivity of lignocellulosic ethanol.

Figure 1 Xylose metabolism in S. cerevisiae.

The metabolic pathways for glucose and xylose metabolism in Saccharomyces cerevisiae. Metabolic pathways for xylose uptake are indicated, including glycolysis and gluconeogenesis, oxidative and non-oxidative pentose phosphate pathway and PK-PTA-AADH Pathway, and the main genes involved in xylose metabolism. Colored boxes represent different paths; names in black represent consumed/produced molecules; the names in gray are the genes that encode the enzymes that participate in each reaction; red are the cofactors; in bold the heterologous genes responsible for key enzymes in xylose metabolism in S. cerevisiae. (XYL1 = xylose reductase), (XYL2 = xylitol dehydrogenase) and (XYLA = xylose isomerase).

In this context, this review focuses on the main approaches used to unravel the genomic structure that is related to this phenotype and the contribution of such genotypes to enhancing xylose metabolism and ethanol production in S. cerevisiae. The main biotechnological strategies used will be addressed, such as deletion of genes that hinder xylose metabolism; overexpression of genes that increase xylose metabolism; gene expression fine tuning for optimized pentose metabolism; improvement in cofactor availability in the oxidoreductive pathway; and expression of optimized transporters to increase xylose assimilation. An overview of the xylose assimilation pathways and the main challenges in the heterologous expression of each of them will also be discussed. Although most results presented here were developed on a lab scale bearing in mind commercial applications, minimum industrial settings have been directly applied in published research, and therefore will not be the focal point in this work.

Survey Methodology

Articles were identified in Google Scholar and in National Center for Biotechnology Information (NCBI) databases for a broader analysis of the results, using the terms: xylose consumption genes, new xylose isomerases, new xylose reductase genes, xylose metabolism, xylose reductase, xylitol dehydrogenase, genetic modifications, metabolic engineering, cofactor preference, targets for deletion, targets for overexpression, xylose transporters, Saccharomyces cerevisiae, Pichia stipitis and ethanol production. After a thorough reading of the articles, those referring to xylose consumption and ethanol production by Saccharomyces cerevisiae were selected. In the process of choosing relevant works, the most recent ones with outstanding results and other pertinent studies in the area were chosen. We tracked articles referring to the first mention in the literature of a certain genetic target and we identified papers that investigated such genes. The analysis allowed us to identify the experimental articles with the highest citations, which were also sorted out. A total of 160 papers were chosen.

Enabling xylose metabolism in S. cerevisiae

The oxidoreductive pathway

The oxidoreductive pathway for xylose consumption is found in fermenting yeast and fungal species, and presents two steps catalyzed by the enzymes xylose reductase (XR) and xylitol dehydrogenase (XDH) (Jeffries, 1983; Ho, Chen & Brainard, 1998). In the first reaction, XR reduces xylose to xylitol, preferentially using NADPH over NADH as a cofactor, in most cases. The xylitol produced is oxidized to xylulose by the enzyme XDH, which naturally uses only NAD+ as a cofactor. The difference in the cofactor preference between the XR and XDH enzymes causes an imbalance that generates xylitol accumulation and consequently reduces ethanol production (Jeffries, 2006). Some studies have also indicated that, although the oxidoreductive pathway can present an imbalance of enzyme cofactor preference, and consequently the accumulation and production of xylitol, it is thermodynamically more favorable than the isomerase pathway, performing faster xylose assimilation in genetically modified strains (Karhumaa et al., 2007b; Bettiga, Hahn-Hägerdal & Gorwa-Grauslund, 2008; Li et al., 2016). Table 1 summarizes the main work that has expressed this pathway in S. cerevisiae.

Table 1 Heterologous pathways for xylose fermentation in S. cerevisiae: the main genes used for the oxidoreductive (XR/XDH) and xylose isomerase (XI) pathways.

Pathway	Gene	Source Microorganism	Codon otimization	Xylose consumed/ Ethanol produced	Yield in g of ethanol per g of substrate	Ethanol yield	Reference	
XR/XDH	XYL1/XYL2	Scheffersomyces stipitis	Y	14 g/L/0.73 g/L	0.052	10%	(Kötter et al., 1990)	
XYL1/XYL2	Scheffersomyces stipitis	N	34 g/L/NP	–	–	(Walfridsson et al., 1995)	
XYL1/XYL2	Scheffersomyces stipitis	Y	50 g/L/∼22.5 g/L	0.45	88%	(Ho, Chen & Brainard, 1998)	
XYL1.2	Spathaspora passalidarum	N	50 g/L/20 g/L	0.4	78%	(Cadete et al., 2016)	
XI	XYLA	Thermus thermophilus	Y	10.4 g/L/1.3 g/L	0.125	24.4%	(Walfridsson et al., 1996)	
XYLA	Clostridium phytofermentans	Y	∼18 g/L /∼7.74 g/L	0.43	84%	(Brat, Boles & Wiedemann, 2009)	
XYLA	Bacteroides stercoris	N	15.7 g/L/4.9 g/L	0.312	66%	(Ha et al., 2011)	
XYLA	Prevotella ruminicola	Y	32.1 g/L/13.6 g/L	0.41	82.9%	(Hector et al., 2013)	
XYLA	Piromyces sp. E2	N	20 g/L/8.68	0.43	84.5%	(Kuyper et al., 2003; Kuyper et al., 2004;
Kuyper et al., 2005)	
XYLA	Piromyces sp. E2	Y	40 g/L/16.8 g/L	0.41	81%	(Zhou et al., 2012)	
XYLA	Orpinomyces sp.	N	15.55 g/L/6.05 g/L	0.39	78%	(Madhavan et al., 2009)	
XYLA	xym1 and xym2 (soil metagenomic library)	N	NM	NM	NM	(Parachin & Gorwa-Grauslund, 2011)	
XYLA (K11T/D220V)	Bovine rumen	N	∼18 g/L/∼7.5 g/L	0.06	80%	(Hou et al., 2016)	
XYLA	Reticulitermes speratus	Y	51 g/L/20 g/L	0.39	77%	(Katahira et al., 2017)	
XYLA	Odontotaenius disjunctus	Y	NM	NM	NM	(Silva et al., 2021)	
Notes.

NP Not produced

NM the value was not measured

The heterologous genes XYL1 and XYL2 from S. stipitis, which encode the enzymes XR and XDH, respectively, have already been used in the construction of a recombinant S. cerevisiae strain expressing the xylose oxidoreductive assimilation pathway (Kötter et al., 1990; Walfridsson et al., 1995; Ho, Chen & Brainard, 1998; Eliasson et al., 2000). Therefore, Kötter et al. (1990) obtained a theoretical ethanol yield of 10%; later, in a study developed by Ho, Chen & Brainard (1998), a theoretical ethanol yield of 88% was obtained. Two genes encoding XR (XYL1.1 and XYL1.2) have also been identified in the genome of Spathaspora passalidarum. The XR encoded by the XYL1.2 gene was cloned and expressed in S. cerevisiae, and the obtained strain presented a higher activity of XR with NADH. Such a feature allowed an efficient consumption of xylose resulting in an ethanol yield of 78%, generating an improvement in ethanol production, as well as a lower xylitol production (Cadete et al., 2016).

Other studies have also used metabolic engineering strategies to reduce the accumulation of xylitol in the xylose metabolization process. Such accumulation is attributed to the excessive increase of NADH unable to be recycled by respiration under oxygen-limited conditions. This condition is related to the difference in cofactor preferences of XR (greater affinity for NADPH than for NADH—converting xylose into xylitol) and XDH (preferably NAD+—converting xylitol into xylulose) and expression levels of these heterologous enzymes in S. cerevisiae (Karhumaa et al., 2007a; Hou et al., 2007; Matsushika & Sawayama, 2008). By changing cofactor affinity through specific amino acids changes in the binding domain of such enzymes, it was possible to reduce unfavorable xylitol excretion during xylose fermentation and optimize the XR-XDH relationship (Watanabe, Kodaki & Makino, 2005; Watanabe et al., 2007a; Hou et al., 2007; Matsushika et al., 2008). In addition, other strategies have relied on modifications to the redox environment in yeast metabolism in order to yield higher cofactor availability and consequently higher rates of ethanol production (Verho et al., 2003; Bro et al., 2006). Strategies for cofactor manipulation will be further discussed in this paper.

The isomerase pathway

The isomerase pathway is mainly found in bacteria, and represents a single-step conversion of xylose to xylulose, catalyzed by the metal ion-dependent enzyme xylose isomerase (XI) (Sarthy et al., 1987; Zhou et al., 2012; Kwak & Jin, 2017). This reaction does not require cofactors, and thus does not exhibit the redox imbalance observed in the oxidoreductive pathway when expressed in S. cerevisiae, also eliminating xylitol overproduction (Sarthy et al., 1987; Kwak & Jin, 2017). On the other hand, xylose isomerase genes (XYLA) are often not functional in S. cerevisiae. Previous efforts to express XI from Escherichia coli (Sarthy et al., 1987), Bacillus subtilis (Amore, Wilhelm & Hollenberg, 1989), Actinoplanes missouriensis (Amore, Wilhelm & Hollenberg, 1989), Thermoanaerobacterium thermosulfurigenes (Moes, Pretorius & Zyl, 1996) and Streptomyces rubiginosus (Gárdonyi & Hahn-Hägerdal, 2003) in S. cerevisiae have not been successful. Nevertheless, the possibility to functionally express heterologous XI (usually with codon-optimization) associated with metabolic flux optimizations and evolutionary engineering in S. cerevisiae has enabled the projection of strains with the ability to ferment xylose at an industrial scale (Demeke et al., 2013). The main XI expressed in S. cerevisiae with confirmed activity in xylose consumption are described in Table 1.

The first functional XI was identified in the bacterium Thermus thermophilus back in 1996: when episomally expressed in S. cerevisiae using the yeast PGK1 promoter and terminator, the recombinant strain was able to produce ethanol equivalent to 24.6% of the theoretical yield. Low ethanol productivity in this strain is linked to the thermophilic enzyme’s low activity at 30 °C (Walfridsson et al., 1996). Other functional bacterial XYLA were found in Clostridium phytofermentans (Brat, Boles & Wiedemann, 2009), Bacteroides stercoris (Ha et al., 2011) and Prevotella ruminicola (Hector et al., 2013). For the first two sequences, strong and constitutive promoters were used for the construction of an expression cassette: HXT7 and TEF1, respectively. C. phytofermentans’s XI optimized for expression in S. cerevisiae allowed an ethanol production corresponding to 84% of the maximum theoretical yield (Brat, Boles & Wiedemann, 2009), while the B. stercoris allele without codon optimization enabled 66% of this value in the engineered strain (Ha et al., 2011). For P. ruminicola, XYLA was codon-optimized and expressed in a high copy plasmid, allowing 68.6% of the ethanol theoretical yield. After an adaptive laboratory evolution (ALE) using serial batch cultures of the transformed strain in a medium with xylose, an increase of 14% in ethanol yield was observed (Hector et al., 2013).

Anaerobic fungi are also source microorganisms for functional XI in S. cerevisiae. Initially, an effective XYLA was identified in Piromyces sp. E2 (ATCC 76762)—isolated from the feces of an Indian elephant (Kuyper et al., 2003). For the expression of this XI, Kuyper et al. (2003) used a vector carrying the XYLA gene from this fungus without codon optimization induced by the constitutive promoter TPI1 (Kuyper et al., 2003; Kuyper et al., 2004). Subsequently, the yeast underwent genetic modifications combined with ALE in xylose to optimize sugar consumption (Kuyper et al., 2004; Kuyper et al., 2005); the evolved strain showed a high rate of ethanol production from xylose (84.5% of the theoretical yield), without xylitol accumulation. Zhou et al. (2012) engineered a S. cerevisiae strain overexpressing a codon-optimized Piromyces sp. E2’s XI under TDH3 promoter, via a multiple copy plasmid. Further genetic modifications and evolutionary engineering rendered ethanol production equivalent to 81% of the theoretical yield. The authors suggested that the high expression level of XYLA was caused by multiple copy genomic integration in the evolved recombinant strain, which contributed to more efficient xylose assimilation.

Orpinomyces sp.—another anaerobic fungus, isolated from bovine rumen fluid—was found to also express the xylose isomerase enzyme, bearing 94% of amino acid sequence identity to Piromyces’ XYLA, and similar specific enzyme activity (Madhavan et al., 2009). In the construction of a recombinant S. cerevisiae strain expressing XI from Orpinomyces sp., the gene was cloned in a high copy vector under the control of GAPDH promoter for episomal expression, and other genetic modifications were introduced to enhance xylose conversion. In this work, Madhavan et al. (2009) reported an ethanol yield equivalent to 78% of the maximum theoretical (Madhavan et al., 2009).

Metagenomics approaches have boosted the identification of new enzymes with xylose isomerase activity. Parachin & Gorwa-Grauslund (2011), reported two new genes encoding functional XI in S. cerevisiae that were isolated from a soil metagenomic library (Parachin & Gorwa-Grauslund, 2011). Degenerated primers and a protein sequence similarity-based screening were applied to identify such genetic information. However, despite being functionally expressed, the aerobic growth rate in xylose of recombinant S. cerevisiae strains carrying multiple copy plasmids expressing such XYLA under the same promoter (TEF1) was much lower compared to the growth of yeast expressing the Piromyces’s XI under the same conditions (Parachin & Gorwa-Grauslund, 2011). In this study, yeasts containing the new XI were able to grow at a rate of 0.02 hour−1 in xylose, while the strain expressing Piromyces’s XYLA grew at 0.07 hour−1. Ethanol production was not assessed for the newly identified XI.

Two other functional XIs were prospected in a metagenomics library from bovine rumen contents (Hou et al., 2016) and from a cDNA library of the protists residing in the hindgut of the termite Reticulitermes speratus (Katahira et al., 2017). The XI obtained from R. speratus was evaluated through episomal expression in S. cerevisiae using the GAP1 promoter, resulting in an ethanol theoretical yield of 77% (Katahira et al., 2017). More recently, metagenomic data derived from resident microorganisms in the gut of the woody beetle Odontotaenius disjunctus revealed a new functional XI (Silva et al., 2021). For that, a methodology that associates direct metagenome reconstruction combined with in vitro gene optimization and synthesis was used. The expression of this new XI in S. cerevisiae resulted in a 50% faster aerobic growth compared to XI from Piromyces sp. on xylose media, while no ethanol production was observed (Silva et al., 2021).

Endogenous xylose metabolism in S. cerevisiae

While wild-type S. cerevisiae strains are not recognized for their xylose-fermenting ability—which foments research on the expression of heterologous pathways—, the ability to grow in small concentrations of pentose has been reported (Toivari et al., 2004; Attfield & Bell, 2006; Wenger, Schwartz & Sherlock, 2010), suggesting the presence of a complete native xylose metabolization pathway. Studies suggest that this phenomenon is possible due to the presence of endogenous genes encoding putative enzymes of the oxidoreductive pathway (XR and XDH). In the genome of laboratory strain S288c, several genes encoding putative enzymes of the xylose pathway or showing a correlative contribution to the xylose consumption phenotype were identified. Genes GRE3, GCY1, YPR1, YDL124W and YJR096W encode putative XR enzymes while XYL2, SOR1 and SOR2 express enzymes with activity homologous to XDH (Wenger, Schwartz & Sherlock, 2010). However, the specific activity of these enzymes is much lower in S. cerevisiae when compared to other xylose-fermenting yeasts (Batt et al., 1986). Therefore, many efforts have been made to understand the role of these enzymes in S. cerevisiae or to identify other genes linked to xylose consumption.

In this sense, Traäff-Bjerre et al. (2004) performed both deletion and overexpression of the endogenous GRE3 to evaluate its contribution to xylose consumption. The gene knockout led to decreased xylitol formation by 49%, while its overexpression under a PGK1 promoter and terminator generated an increment in ethanol production by 116% in a recombinant strain expressing XDH from S. stipitis (Traäff-Bjerre et al., 2004). Toivari et al. (2004) enabled growth of S. cerevisiae in xylose in a medium containing glucose in the presence of oxygen by overexpressing endogenous GRE3 and XYL2. However, the mutant strains presented slower growth and greater xylitol accumulation compared to a recombinant S. cerevisiae strain expressing XR and XDH from S. stipitis.

Wenger, Schwartz & Sherlock (2010) described the endogenous gene XDH1, encoding a putative XDH, as responsible for enabling xylose consumption in an S. cerevisiae wine strain. Through mass segregation analysis (BSA) and yeast tiling arrays using the xylose-consuming wine strain and a non-consuming laboratory strain (S288C), the authors identified that the positive phenotype for xylose consumption is linked to a unique, dominant locus, located in a subtelomeric region on the right-end of chromosome XV—not present in the genome of S288C. Confirmation of the contribution of the XDH1 gene to the xylose consumption phenotype was accomplished by its deletion in the S. cerevisiae wine strain, after which the phenotype was nullified. The gene was also cloned and expressed episomally in the laboratory yeast S288C, endowing this yeast with xylose consumption ability, the phenotype being lost upon plasmid removal.

Furthermore, in the same study (Wenger, Schwartz & Sherlock, 2010), other genes and their correlation with the positive xylose metabolism phenotype in S. cerevisiae were analyzed by performing different knockout combinations. By deleting each XDH (sor1Δ, sor2Δ, xyl2Δ) separately, an improvement in xylose consumption was observed, while deleting the three genes at the same time resulted in an enhanced phenotype, suggesting that endogenous XDHs may limit the xylose-consuming ability of non-recombinant S. cerevisiae. Wenger, Schwartz & Sherlock (2010) also confirmed the contribution of two putative XR genes (GRE3 and YPR1) to growth on xylose: the two genes were the only ones that contributed significantly to the ability to utilize xylose in the used background, GRE3 being the one that most affected the phenotype. A mutant presenting a gre3Δypr1Δ genotype had its xylose consumption phenotype almost completely removed, indicating that the presence of these genes allows the metabolism of this pentose in S. cerevisiae.

Rewiring Metabolic Pathways

Even though genes encoding enzymes of the oxidoreductive pathway, as well as an active xylitol dehydrogenase, are found in S. cerevisiae, this yeast does not efficiently consume xylose. Therefore, overexpression of endogenous genes and/or insertion of heterologous enzymes (i.e., XR, XDH or XI) are common strategies applied to use this microbe as a platform for xylose assimilation, as previously discussed. Nevertheless, additional modifications are required to optimize the metabolic flux of this sugar, especially for commercial purposes. In this manner, different approaches have been used to optimize the metabolic pathways (Fig. 1) to increase cellulosic ethanol yield. The main strategies are: (I) knock-out of genes that hinder the flux of xylose metabolism; (II) overexpression of genes that can increase xylose metabolism; (III) use of specific promoters and terminators for gene expression fine-tuning; (IV) improvement of cofactor availability for the XR-XDH pathway; and (V) expression of transporters with higher affinity towards xylose to increase sugar assimilation. These different strategies will be discussed in detail in the following sections.

Deletions

Several approaches have been explored to identify genes related to xylose consumption in S. cerevisiae, such as (I) reverse engineering (Bengtsson et al., 2008; Verhoeven et al., 2017; Tran Nguyen Hoang et al., 2018); (II) genome-wide synthetic genetic array (SGA) screens (Usher et al., 2011); (III) transposon mutagenesis (Ni, Laplaza & Jeffries, 2007); and (IV) omics approaches for comparative analysis of mutated or evolved xylose-fermenting strains and their respective parents (Kim et al., 2013; Sato et al., 2016; dos Santos et al., 2016; Palermo et al., 2021). Within these studies, several genes were suggested as knockout targets that either directly contribute to xylose metabolism or that, associated with other deletion/superexpression gene targets, enhance this phenotype in S. cerevisiae. Following, we present the main gene deletions described in the literature that are beneficial for ethanol production from xylose in S. cerevisiae. This information is summarized in Table 2.

Table 2 Deletion targets that contribute to improved xylose metabolism in S. cerevisiae.

Deletion targets	Relevant genetic background	Reported phenotype improvement	Reference	
GRE3	Thermus thermophilus XYLA/XKS1	Xylitol formation decreased two-fold, and which produced ethanol from xylose with a yield of 0.28 mmol	(Träff et al., 2001)	
Thermus thermophilus mutated XYLA / XKS1	Deletion of GRE3 was crucial for ethanol production as reduction of xylitol formation was observed	(Lönn et al., 2003)	
NFG1 / MNI1 / RPA49	Scheffersomyces stipitis XYL1 and XYL2 / XKS1	Improved growth rates on xylose in aerobiosis compared to the reference strain: 173% (nfg1Δ), 62% (mni1Δ) and 90% (rpa49Δ) faster	(Bengtsson et al., 2008)	
NFG1	XYL1 / XYL2 / XKS1	Improvement of xylose consumption at low concentrations and in co-fermentation of glucose and xylose; deletion of NFG1 improved aerobic growth on xylose	(Parachin et al., 2010)	
BUD21 / ALP1 / ISC1 / RPL20B	XYLA / XKS1	Individual deletion of the 4 genes improved xylose assimilation in 27.6% (bud21Δ), 15.5% (alp1Δ), 22.4% (isc1Δ) and 12.1% (rpl20bΔ); production of ethanol in bud21Δ cells even without the presence XYLA	(Usher et al., 2011)	
PMR1	Piromyces sp. XYLA / XKS1 / RKI1 / RPE1 / TKL1 / TKL2 / TAL1 / NQM1 / gre3Δ	Deletion of PMR1 allowed anaerobic growth on xylose	(Verhoeven et al., 2017)	
PMR1 / ASC1	Piromyces mutated XYLA3* / TAL1 / XKS1 / gre3Δ / pho13Δ	Mutated PMR1 and ASC1 consumed 114.8% and 59.6% more xylose in relation to the control, respectively	(Tran Nguyen Hoang et al., 2018)	
GRE3 / HOG1 / IRA2 / ISU1	Clostridium phytofermentans XYLA / TAL1 / S. stipitis XYL3	The mutation in IRA2 only affects anaerobic xylose consumption; loss of ISU1 function is indispensable for anaerobic xylose fermentation; Faster conversion of xylose obtained by deleting the gre3Δ, hog1Δ, ira2Δ and isu1Δ genes simultaneously	(Sato et al., 2016)	
ISU1 / SSK2	Orpinomyces sp. XYLA / XKS1 / RKI1 / RPE1 / TKL1 / TKL2 / TAL1 / gre3Δ	ISU1 or SSK2 null strains showed improvement in xylose metabolism in unevolved yeast cells	(dos Santos et al., 2016)	
PHO13	Scheffersomyces stipitis XYL1 and XYL2	Improvement in xylose assimilation	(Ni, Laplaza & Jeffries, 2007)	
Scheffersomyces stipitis XYL1, XYL2 and XYL3	Upregulation of the enzymes from PPP and NADPH-producing enzymes; improved xylose metabolism	(Kim et al., 2015)	
Scheffersomyces stipitis XYL1, XYL2 and XYL3	Transcriptional activation of genes from PPP; 98% reduction of sedoheptulose by upregulation of tal1 in mutant strains (pho13Δ)	(Xu et al., 2016)	
PHO13 / ALD6	Scheffersomyces stipitis XYL1 and XYL2	pho13Δ strains presented a shorter lag time using xylose as carbon source and showed an improved xylose fermentation / ald6Δ strains showed improvement in the efficiency of xylose fermentation and prevention of acetate accumulation	(Kim et al., 2013)	
GCR2	Scheffersomyces stipitis XYL1, XYL2 and XYL3	gcr2Δ cells with better xylose utilization and ethanol production.	(Shin et al., 2021)	
THI2	Ru-XYLA (where Ru represents the rumen bovine) / XKS1 / RKI1 / RPE1 / TKL1 / TKL2 / TAL1 / cox4Δ / gre3Δ	Deletion increases 17.4% in growth rate, increase of 26.8% in specific xylose utilization rate and 32.4% increase in specific ethanol production rate in co-fermentation of glucose and xylose	(Wei et al., 2018)	
CCC1 / BSD2	Orpinomyces sp. XYLA / XKS1 / RKI1 / RPE1 / TKL1 / TKL2 / TAL1 / gre3Δ	ccc1Δ and bsd2Δ strains had a 9-fold and 2.3-fold increase in xylose consumption	(Palermo et al., 2021)	
HAP4	Scheffersomyces stipitis XYL1, XYL2 and XYL3	hap4Δ strain: 1.8-fold increase in ethanol production from xylose; production of 10.38 g/L of ethanol; ethanol yield of 0.41 g/g of xylose	(Dzanaeva et al., 2021)	

The GRE3 gene encodes a non-specific aldose reductase that functions as an NADPH-dependent XR and consequently contributes to the formation of xylitol (Traäff-Bjerre et al., 2004). Therefore, the deletion of this gene is paramount for improved ethanol yield from xylose when using the isomerase pathway (Träff et al., 2001; Lönn et al., 2003; Karhumaa, Hahn-Hägerdal & Gorwa-Grauslund, 2005). For instance, compared to a GRE3 strain, gre3Δ were able to reduce xylitol production by 50%, boosting ethanol yield. In addition, most of the XIs expressed in S. cerevisiae strains were sensitive to the presence of xylitol—indicating that this metabolite can act as a potent inhibitor of these enzymes (Yamanaka, 1969; Lönn et al., 2003). However, it is noteworthy that GRE3 knockout was also related to reduction in biomass production, suggesting that the fine-tuning of gene expression would be preferable to deletion (Traäff-Bjerre et al., 2004).

In order to identify new gene targets for improved xylose metabolism, Bengtsson et al. (2008) compared strains with varying degrees of this phenotype using a genome-wide transcription analysis, and further reverse genetic engineering. Strains with null NFG1 (negative regulator of the filamentous growth MAPK pathway); MNI1 (methyltransferase), or RPA49 (RNA polymerase) showed growth on xylose 173%, 62% and 90% times better, respectively, compared to the reference strains. These results suggested that NFG1, MNI1, and RPA49 could be involved in central carbon metabolism and xylose utilization in S. cerevisiae (Bengtsson et al., 2008). Later, the positive effect of NFG1 deletion on xylose fermentation was also confirmed in another study (Parachin et al., 2010). The phenotype for nfg1Δ cells included different assimilation of other sugars and increased xylitol production, suggesting that NFG1 is related to sugar transport or signaling. In general, strains with an NFG1 knockout genotype were able to consume 27.1% of available xylose, while the reference yeast consumed only 18% of the sugar (Parachin et al., 2010).

Meanwhile, Usher et al. (2011) used a genome-wide synthetic genetic array (SGA) screening methodology to identify deletion mutants and evaluate the contribution of non-essential genes to xylose utilization in a recombinant S. cerevisiae (strain expressing xylA from Piromyces sp. E2). Four deletion mutants were identified: BUD21 (component of the small ribosomal subunit, SSU, processome), ALP1 (arginine transporter), ISC1 (inositol phospho-sphingolipid phospholipase C) and RPL20B (component of the large ribosomal subunit, 60S). In order to evaluate the influence of each gene on the phenotype, they were individually knocked out, confirming that all contribute positively to xylose consumption. Xylose consumption improved 27.6%, 15.5%, 22.4%, and 12.1%, respectively, for each deleted gene compared to the reference strain. The authors suggested that such genes are xylose metabolic suppressors and could be regulators at the transcriptional or translational level. BUD21 is of particular interest, as its exclusion allows certain aspects of the stress response not to be activated, making it possible to circumvent some of the initial stress conditions that occur during xylose fermentation in S. cerevisiae. Despite the improvement obtained in the consumption of xylose, such genes may have a synergistic relationship with the response to stress, indicating the need for further study to assess the impact on the robustness of yeasts used in the fermentation of lignocellulosic hydrolysates.

The mutations G249V and G1161A in PMR1, a gene responsible for encoding a Golgi Ca2+/Mg2+ ATPase, was identified by Verhoeven et al. (2017) in an S. cerevisiae strain expressing Piromyces E2′XI and other additional modifications (overexpression of XKS1, RKI1, RPE1, TKL1, TKL2, TAL1, NQM1 and gre3Δ) after ALE in an anaerobic culture (Verhoeven et al., 2017). In parallel, Tran Nguyen Hoang et al. (2018) reported another mutation (G681A) in PMR1, found in an evolved recombinant S. cerevisiae strain harboring a mutant xylose isomerase gene from Piromyces sp. (XYLA *3) and other additional metabolic alterations (overexpression of XKS1 and TAL1 and, gre3Δ and pho13Δ). To understand whether both mutations were accompanied by loss of function, a PMR1 deletion was performed by both authors, which allowed phenotype improvement regarding ethanol production from xylose. The authors suggested that negative regulation of PMR1 expression leads to the accumulation of manganese ions inside the cell, which would be available for ion-dependent enzymes such as xylose isomerases (Tran Nguyen Hoang et al., 2018). In general, there was an improvement of 114.8% in consumed xylose and 195.9% in ethanol production, in relation to the strain containing the original gene.

In that same study, Tran Nguyen Hoang et al. (2018) also described a mutation (Q237*) in ASC1, encoding the beta subunit of the G protein and the guanine dissociation inhibitor for Gpa2p. This gene is known as a negative regulator of several metabolic and signal transduction pathways. When the mutated gene was deleted, as well as when the mutation was expressed in knockout strains, a significant improvement of 59.6% in xylose consumption and 104.4% in ethanol production was observed. Therefore, the authors concluded that the Q237* mutation in ASC1 is correlated with the loss of function of that gene (Tran Nguyen Hoang et al., 2018). ASC1 had already been associated with cell growth in oxygen-limited conditions and, when deleted, with the overexpression of genes correlated with xylose metabolism, being a negative regulator of metabolic pathways and of signal transduction. In particular, ASC1 acts on the repression of the transcription factor (TF) GCN4, responsible for the regulation of genes linked to xylose metabolism in strains that have a high fermentative profile (Tran Nguyen Hoang et al., 2018).

Sato et al. (2016) described null genotypes—including epistatic interactions—that alter the metabolic regulation of S. cerevisiae and enhance anaerobic xylose consumption, when analyzing the genome sequencing of a strain (genotype: TAL1 overexpression, XYL3 from S. stipitis and XYLA from C. phytofermentans) that underwent ALE in lignocellulosic hydrolyzate (Parreiras et al., 2014). Mutations G136A, A844del, G8782T, C412T were observed in the genes GRE3, HOG1 (a component of MAP kinase, MAPK, signaling), IRA2 (a GTPase activating protein) and ISU1 (a scaffolding protein involved in mitochondrial iron-sulfur cluster assembly), respectively. For validation, the four genes underwent deletion in a combined manner in different yeast strains, resulting in faster anaerobic xylose consumption regardless of the background. It was suggested that the loss of function of ISU1 is indispensable for the anaerobic fermentation of xylose, as well as epistatic interactions with mutations in IRA2, HOG1 and GRE3. Null ISU1 mutants were able to consume about 75% more xylose under aerobic conditions, and combined with the deletions of IRA2, HOG1 and GRE3 resulted in increased xylose-specific consumption and ethanol production rates comparable to the phenotype of the evolved strain. The authors claim that these deleterious genetic alterations influenced different metabolic pathways, such as xylose catabolism, the pentose phosphate pathway (PPP), the glycolytic pathway and aerobic respiration. Such changes together resulted in increased aerobic consumption and anaerobic fermentation of xylose (Sato et al., 2016).

In the work conducted by dos Santos et al. (2016), two mutations that improve xylose consumption were identified. In this study, an industrial S. cerevisiae strain was modified (gre3Δ, Orpinomyces sp XI and overexpression of XKS1, RKI1, RPE1, TKL1, TKL2 and TAL1) for pentose metabolism, associated with ALE in xylose. Genetic mapping of the evolved strains revealed that ISU1 harbors mutations in some isolates, whereas SSK2 (a member of the MAPKKK signaling pathway) presented polymorphisms in others. For phenotype validation, the authors created knockout strains for both genes, resulting in an improvement in xylose metabolization compared to the wild-type strain. SSK2 deletion in the non-evolved parental strain resulted in an 80% increase in fermentation efficiency. The deletion of ISU1 allowed a reduction in fermentation time, from 80 to 40 h in the evolved lineage (representing an upgrade in fermentation efficiency of 86% for isu1Δ cells). This is similar to that found in the strains where the mutations were identified, indicating that these genetic changes led to gene inactivation (dos Santos et al., 2016).

One gene that has been extensively investigated is PHO13, encoding a phosphatase with specific dephosphorylating activity on two side-products of central carbohydrate metabolism. This gene has been the deletion target of different inquiries, in order to understand its influence on xylose metabolism in S. cerevisiae (Van Vleet, Jeffries & Olsson, 2008; Fujitomi et al., 2012; Li et al., 2014; Lee, Jellison & Alper, 2014; Bamba, Hasunuma & Kondo, 2016). Loss of function mutations in PHO13 in recombinant strains have been identified in different studies, regardless of the initial xylose uptake pathway (Ni, Laplaza & Jeffries, 2007; Kim et al., 2013). Insertional transposon mutagenesis was used to identify that PHO13 deletion increased transcripts for TAL1, indicating that overexpression of transcripts for downstream enzymes of the xylose pathway may improve the assimilation of this sugar (Ni, Laplaza & Jeffries, 2007). Through a metabolomic analysis, it was revealed that the positive regulation of TAL1, which prevents sedoheptulose accumulation, is the critical point for improved xylose metabolism in pho13Δ mutant S. cerevisiae strains (Xu et al., 2016). In the same fashion, it was suggested that knockout of PHO13 results in transcriptional and metabolic changes favorable for xylose fermentation, in particular, transcriptional activation of PPP genes and NADPH-producing enzymes as part of an oxidative stress response mediated by Stb5 activation (Kim et al., 2015). In another study, it was indicated that loss of PHO13 function, acquired after ALE in xylose, plays an important role in improving xylose consumption rates and ethanol yields (Kim et al., 2013).

On the other hand, a recent study by Shin et al. (2021) reported that the phenotype for xylose metabolization had not been affected by PHO13 deactivation in S. cerevisiae strains. Through resequencing of the pho13Δ strains, a loss-of-function Glu204* mutation in GCR2 was identified and indicated as responsible for the improvement in the xylose consumption phenotype. GCR2 is a global TF correlated with glucose metabolism. Deletion of GCR2 led to positive regulation of the PPP genes, as well as negative regulation of glycolytic genes, with the changes being more significant under xylose conditions than in the presence of glucose. Although no synergistic effect was found between the deletion of PHO13 and GCR2 in improving xylose fermentation, GCR2 was indicated as a knockout target to enhance ethanol production.

Many other genes were also identified and suggested as deletion targets to improve xylose fermentation in S. cerevisiae, directly or indirectly, alone or associated with other modifications (i.e., combined deletions or overexpression). Deletion of ALD6—encoding a NADPH-dependent aldehyde dehydrogenase, part of the central carbon metabolism—yielded an improvement in xylose fermentation efficiency (Kim et al., 2013). In 2018, Wei and colleagues (Wei et al., 2018) suggested that deletion of the TF THI2 (activator of thiamine biosynthetic genes) enables the co-fermentation of glucose and xylose by increasing ribosome synthesis, generating an increase in the specific utilization rate of xylose by 26.8%. Palermo et al. (2021), meanwhile, analyzed the effect of metal homeostasis under xylose fermentation and suggested two new deletion targets for metabolic engineering of S. cerevisiae: CCC1 (vacuolar Fe2+/Mn2+ transporter) and BSD2 (protein involved in heavy metal ion homeostasis) (Palermo et al., 2021). More recently, interruption of transcription factors of xylose catabolism (ZNF1, SIP4, ADR1, TUP1 and HAP4) were evaluated in a xylose-fermenting S. cerevisiae strain; however only deletion of hap4Δ (global regulator of respiratory gene expression) generated an increase in ethanol production from xylose compared to the parental strain (Dzanaeva et al., 2021).

Overexpression

Among the rational genetic modifications performed in yeast to improve xylose fermentation, gene overexpression has become a prominent strategy, because it directly contributes to accelerating the uptake of this pentose and increases xylose metabolism flux in genetically modified S. cerevisiae strains (Nevoigt, 2008). Therefore, in this section the main overexpression targets in S. cerevisiae aiming at an optimized xylose consumption will be highlighted. The summarized information can be found in Table 3.

Table 3 Overexpression targets that contribute to improved xylose metabolism in S. cerevisiae.

Overexpression targets	Relevant genetic background	Reported phenotype improvement	Reference	
XKS1	Scheffersomyces stipitis XYL1, XYL2	Fermentation at high xylose concentrations and reduced xylitol production	(Ho, Chen & Brainard, 1998)	
Scheffersomyces stipitis XYL1, XYL2 and XYL3 / pho13Δ	Overexpression of XK genes (XYL3 or XKS1) increases the rate of xylose assimilation and maintain ATP levels inside cells	(Kim et al., 2013)	
LSK1 - xylulokinase mutant	Increased enzyme activity improved xylulose conversion and accelerated ethanol production by 30–130%	(Deng & Ho, 1990)	
Scheffersomyces stipitis XYL1, XYL2	Deleterious effect associated with uncontrolled overexpression of XKS1 / xylulose-5-phosphate accumulation and ATP depletion	(Johansson et al., 2001)	
FY1679 (ura3- 52/ura3-52; his3v200/his3; leu2v1/leu2; trp1v63/ trp1; gal2/gal2)	High levels of expression of this gene have a deleterious effect	(Rodriguez-Peña et al., 1998)	
XKS1 / Scheffersomyces stipitis XYL3	Scheffersomyces stipitis XYL1, XYL2	Growth inhibition on xylose / expression levels should consider the metabolic capacity of the strain	(Jin et al., 2003)	
RPE1 / RKI1 / TAL1 / TKL1	Piromyces sp. E2 XYLA / gre3Δ	Specific xylose consumption rate of 1.1 g g−1 h−1	(Kuyper et al., 2005)	
Scheffersomyces stipitis XYL1 and XYL2 / XKS1 / gre3Δ	Increased rate of xylose consumption	(Karhumaa et al., 2007a)	
TAL1	Scheffersomyces stipitis XYL1, XYL2 and XYL3 / gre3Δ	Improvement in xylose assimilation	(Ni, Laplaza & Jeffries, 2007)	
TAL1 / TKL1	Scheffersomyces stipitis XYL1 and XYL2 / XKS1	Important role in xylose consumption and fermentation	(Matsushika et al., 2012)	
XKS1 / TAL1/ TKL1 / SOL3 / GND1	Scheffersomyces stipitis XYL1 and XYL2 / XKS1	Increased consumption of xylose by 31%	(Wahlbom et al., 2003)	
GND1 / SOL3 / TAL1 / RKI1 / TKL1	Orpinomyces sp. XYLA / XKS1 / gre3Δ/ pho13Δ	Improves xylose consumption rate	(Bamba, Hasunuma & Kondo, 2016)	
SOL3 / TAL1	Scheffersomyces stipitis XYL1 and XYL2 / XKS1	Fastest growth on xylose by 19% (SOL3) and 24% (TAL1)	(Bengtsson et al., 2008)	
RPE1	Piromyces sp. XYLA *3 / pho13Δ / gre3Δ/ asc1Δ	Significantly improved xylose utilization	(Hoang Nguyen Tran et al., 2020)	
NRM1/YHP1	Ru-XYLA / XKS1 / RKI1 / RPE1 / TKL1 / TKL2 / TAL1 / cox4Δ/ gre3Δ	NRM1 increased the xylose utilization rate by 30%. YHP1 increased the volumetric xylose utilization rate by 5.6%	(Wei et al., 2018)	
STT4 / RGI2 / TFC3	Ru-XYLA / XKS1 / RKI1 / RPE1 / TKL1 / TKL2 / TAL1 / cox4Δ/gre3Δ	Increased xylose specific utilization rates: STT4 (36.9%), RGI2 (29.7%) and TFC3 (42.8%)	(Wei et al., 2019)	
Piromyces sp. E2 XYLA	Scheffersomyces stipitis XYL3 and TAL1 / RPE1 / RKI1 / TKL1	Xylose consumption rate of 1.866 g g−1 h−1	(Zhou et al., 2012)	
XYL2	Scheffersomyces stipitis XYL1 and XYL3	Increased ethanol yields and decrease in xylitol production	(Kim et al., 2012)	

In both xylose assimilation pathways, xylulose is converted to xylulose-5-phosphate by an endogenous xylulokinase (XK) encoded by XKS1, driving carbon flux to the PPP (Fig. 1). Because XK presents a low activity level, it may limit xylose fermentation, making XKS1 a major target for overexpression. Many studies have evaluated endogenous and exogenous overexpression of XKS1, suggesting that this genetic modification is responsible for a remarkable improvement in xylose fermentation (Deng & Ho, 1990; Ho, Chen & Brainard, 1998; Kim et al., 2013). The first recombinant S. cerevisiae strain overexpressing XKS1 with Pichia stipitis XR and XDH resulted in increased ethanol production and reduced xylitol excretion (Ho, Chen & Brainard, 1998). Meanwhile, some studies have indicated that high XK activity can be harmful to xylose metabolism, inhibiting or reducing xylose consumption—even in cases where improvement in ethanol yield was achieved (Johansson et al., 2001; Jin et al., 2003). Rodriguez-Peña et al. (1998) and Johansson et al. (2001) even associated a deleterious effect in strains with uncontrolled overexpression of XKS1. Despite the controversies, there is agreement on the need for modulated XK expression to obtain efficient xylose fermentation in S. Cerevisiae, especially considering the intrinsic characteristics of the host strain (Jin et al., 2003).

Other overexpression targets that have been described to improve xylose metabolism are the genes responsible for encoding enzymes of the non-oxidative PPP in S. cerevisiae –RPE1, RKI1, TAL1 and TKL1 (Fig. 1). Studies analyzing the bottlenecks for xylose consumption argue that the expression levels of such enzymes are preeminent in the xylose utilization rate in fermenting yeast (Matsushika et al., 2012; Bamba, Hasunuma & Kondo, 2016). Overexpression of TAL1 alone is correlated to an improved xylose assimilation rate in S. cerevisiae (Ni, Laplaza & Jeffries, 2007). In other studies, all genes participating in the non-oxidative PPP, including XKS1, were overexpressed simultaneously, which resulted in improved ethanol production in recombinant S. cerevisiae (Kuyper et al., 2005; Karhumaa et al., 2007a). In a characterization study of the enzymes of the non-oxidative PPP, the effects of the TAL1 and TKL1 genes were analyzed by deletion. It was suggested that the enzymatic activities of the transaldolase and transketolase encoded by these genes, respectively, are limiting for efficient xylose utilization. Furthermore, their overexpression is responsible for an increased flux from the PPP to the glycolytic pathway in recombinant S. cerevisiae (Matsushika et al., 2012).

A molecular analysis of a recombinant xylose-consuming S. cerevisiae strain (expressing the XR-XDH pathway) and its mutant obtained through chemical mutagenesis with ethyl methanesulfonate to improve the ability to metabolize xylose, allowed the identification of different gene targets for overexpression (Wahlbom et al., 2003). Besides those already described, XKS1, TAL1 and TKL1, SOL3 (6-phosphogluconolactonase) and GND1 (6-phosphogluconate dehydrogenase) were also evaluated. The authors reported an improvement in growth rate and xylose uptake when SOL3 and GND1 are overexpressed, attributing it to the altered expression of one or more transcriptional regulators that influence these genes (Wahlbom et al., 2003). Bengtsson et al. (2008), in a similar study, performed a transcriptome analysis of S. cerevisiae strains (expressing S. stipitis XR-XDH) with increased xylose consumption phenotype, in order to identify new targets for metabolic engineering. The authors validated the overexpression of SOL3 and TAL1, in multicopy plasmids, resulting in 19% and 24% in growth improvement, respectively.

In a recent study focusing on the simultaneous co-fermentation of glucose and xylose, the RPE1 gene (responsible for catalyzing a reaction in the non-oxidative part of the PPP) was selected as a target for overexpression, leading to an increased xylose consumption and ethanol production rate. Such results were attributed to a possible metabolic rearrangement of the xylose pathway, due to a cofactor-neutral xylose isomerase mutant present in this recombinant yeast (Hoang Nguyen Tran et al., 2020).

Wei et al. (2018) reported a beneficial effect on xylose metabolism through overexpression of NRM1 (Transcriptional co-repressor of MBF-regulated gene expression) and/or YHP1 (Homeobox transcriptional repressor) in S. cerevisiae. NRM1 overexpression increased the specific xylose use rate by 30.0%, while YHP1 increased the volumetric xylose use rate by 5.6%. The authors suggested that these modifications induced an acceleration in the yeast cell cycles, however, it is still unclear how such factors are affecting xylose metabolism.

In another study by Wei et al. (2019), in order to assess how TF Thi2 affects xylose metabolism, a transcriptomic analysis between a thi2Δ strain and its parent in the glucose depletion and glucose-xylose co-fermentation steps was performed, allowing the identification of new genes correlated with xylose metabolism. Through overexpression, it was indicated that the TFs STT4 (Phosphatidylinositol-4-kinase), RGI2 (respiratory growth induced, function unknown) and TFC3 (subunit of RNA polymerase III transcription initiation factor complex) allowed an increase of specific xylose uptake rate in the strains by 36.9%, 29.7%, and 42.8%, respectively, in the glucose depletion step, allowing glucose-xylose co-fermentation in S. cerevisiae.

In addition to the endogenous overexpression targets, elevated expression of the initial genes of the xylose assimilation pathways—XYL1, XYL2, and/or XYLA—also contributes to more efficient metabolism of this sugar, and has been described as paramount for efficient xylose fermentation in S. cerevisiae (Kim et al., 2012; Zhou et al., 2012). Overexpression of sugar transporters is another interesting approach to improve the performance of xylose-consuming strains, enabling even more effective xylose transport in recombinant S. cerevisiae strains (Tanino et al., 2012). The topic of sugar transporters will be further discussed in this review.

Regulation fine tuning

Metabolic engineering approaches often require fine-tuning gene expression to optimize the activity of certain enzymes and regulatory proteins. The modulation of gene transcription levels is of prime importance to balance metabolic fluxes and increase the production of metabolites of interest (Xu et al., 2021). In S. cerevisiae, promoters are responsible for controlling gene expression programs in response to a variety of circumstances (Maya et al., 2008). However, genes participating in the same metabolic pathway might present different levels of expression, as well as different catabolic intermediates. In this context, a widely used strategy in optimizing metabolic flux is assembling expression cassettes using promoters with different activity levels to fine-tune the metabolic pathway in question (Hubmann, Thevelein & Nevoigt, 2014).

Endogenous S. cerevisiae promoters differ by strength magnitudes (i.e., rates of transcription initiation) and regulation, and are classified as constitutive or inducible. Constitutive promoters have stable expression rates and are constantly active in the cell (Da Silva & Srikrishnan, 2012; Tang et al., 2020). Inducible promoters, on the other hand, are activated in response to different stimuli (Li et al., 2006; Weinhandl et al., 2014). Promoters can be obtained by characterizing gene expression or with targeted modifications in the sequence of already known promoters. The latter can be performed by either increasing transcriptional activation with the addition of upstream activating sequences (UASs) or by altering sequences using random mutations, deletions, nucleosome removal or intron insertion. Error-prone PCR (Feng & Marchisio, 2021), for instance, is a strategy used to obtain promoters with different activity degrees, due to mutations added to it during amplification (Alper et al., 2005).

Jeppsson et al. (2002), in an attempt to equilibrate cofactor imbalance in a strain expressing the XR-XDH pathway, indicated that interruption of the ZWF1 gene (Glucose-6-phosphate dehydrogenase, G6PDH) increased ethanol and decreased xylitol yields (more on cofactor engineering will be reviewed in the next section). However, the strain showed a significant reduction in the xylose consumption rate, indicating the need for fine adjustment of this gene expression. In a new investigation, Jeppsson et al. (2003) used a synthetic promoter library to study the influence of different levels of G6PDH activity on xylose fermentation. Downregulation of ZWF1 using the synthetic promoter YRP13 resulted in the lowest G6PDH activity, which enabled a xylose consumption rate five times faster than the zwf1Δ strain, accompanied by higher ethanol and lower xylitol yields.

In order to optimize xylose fermentation, Lu & Jeffries (2007) developed a multiple-gene-promoter shuffling (MGPS) technique to identify optimal expression levels of genes of interest induced by different promoters in S. cerevisiae. In this study, the genes TAL1, TKL1 and PYK1 (pyruvate kinase) were overexpressed in a recombinant xylose-fermenting S. cerevisiae, expressing the oxidoreductive pathway, under control of the weak GND2 and HXK2 promoters. Such promoters were selected to avoid systemic saturation and obtain a balanced flux of metabolites. The authors describe that the optimum scenario for metabolic engineering was the combination of the GND2 promoter overexpressing TAL1 and the HXK2 promoter overexpressing TKL1 and PYK1. Overall, the study states that balanced overexpression of such genes optimized ethanol production from xylose in S. cerevisiae.

Zha et al. (2012) reported a combined strategy of chassis selection and fine-tuning in the expression of XYL1 and mutated XYL2 to obtain efficient S. cerevisiae strains for xylose fermentation. In the engineered strain, promoters PGK1, ADH1 and truncated ADH1 were used to modulate the expression levels of XYL1, while XYL2 was overexpressed under promoter PGK1 in a multicopy plasmid. The authors concluded that only the strongest promoter (PGK1) was able to improve XR activity, increasing by a factor of 1.7 the ability to assimilate and metabolize xylose. Overexpression of XYL2 allowed for 21% lower xylitol production and 35–40% higher ethanol production.

More recently, Hector & Mertens (2017) suggested the need for regulation at the transcriptional and post-translational levels in S. cerevisiae strains engineered to metabolize xylose. In this study, xylose-regulated synthetic hybrid promoters were developed from the Ashbya gossypii TEF constitutive promoter, a mutation being inserted in the second TATA sequence present at position -63. Furthermore, to control transcription in S. cerevisiae, the xylose-dependent DNA repressor obtained from Caulobacter crescentus was also used. The TEF-xylO2-1 promoter in the presence of xylose showed activity comparable to other known S. cerevisiae promoters, with an increase in activity of up to 25 times in the presence of xylose, revealing an important strategy for further metabolic engineering.

Nambu-Nishida et al. (2018), evaluated 30 S. cerevisiae promoters showing different expression levels, selected through microarray data, in a xylose-metabolizing yeast strain (expressing the XR-XDH pathway and XKS1 overexpression). In the study, it was suggested that TDH3, FBA1 and TDH1 promoters showed high expression in aerobic culture and moderate expression in microaerobic fermentation, while promoters SED1, HXT7, PDC1, TEF1, TPI1 and PGK1 had medium-high expression in the same conditions.

The activities of different native promoters and the synthetic hybrid promoter p3xC-TEF1 (based on the TEF1 promoter core with insertion of three tandem upstream activation sequences of the CLB2 promoter) were evaluated in a S. cerevisiae strain expressing the XR-XDH pathway through a fluorescent reporter protein in the presence of xylose (Xiong et al., 2018). The TDH3 promoter showed the highest activity in the presence of xylose as the only carbon source, followed by the synthetic hybrid (p3xC- tef1) and the TEF1 promoter. In another study, the TDH3 promoter and the CYC1 terminator were used to control expression of the PPP genes (TAL1, TKL1, RKI1 and RPE1), and as a result, improved xylose metabolism was obtained (Kobayashi et al., 2018).

Studies have also focused on the role of terminators and how their transcription regulation interferes with an enhanced metabolic flux (Curran et al., 2013; Matsuyama, 2019). There are two events related to gene expression termination: I) transcriptional termination and II) post-transcriptional regulation. In the first event, the terminator is responsible for determining where the mRNA will be cleaved for the addition of the poly(A) tail; while the second determines the stability, translation efficiency and position of the mRNA (Guo & Sherman, 1996; Tuller, Ruppin & Kupiec, 2009; Mischo & Proudfoot, 2013; Yamanishi et al., 2013; Curran et al., 2015). Curran et al. (2013) (Curran et al., 2013) analyzed 30 gene terminators and obtained S. cerevisiae strains presenting better growth on xylose when the XYLA gene was combined with the TDH3 promoter and CPS1 terminator. The authors indicated that there was an increase in transcriptional levels and, consequently, an increase in the xylose growth rate. This scenario suggested that a strong promoter combined with a weak terminator can increase metabolic flux, with terminators also being responsible for modulating protein expression. Finally, it was suggested that a high-expression terminator combined with a weaker promoter could achieve results equivalent to those obtained by strong promoters paired with standard terminators.

Cofactors

As previously stated, although xylose fermentation by S. cerevisiae expressing heterologous XR and XDH is possible, the resulting strains present low ethanol productivity while accumulating a considerable amount of xylitol. Xylitol production is mainly attributed to the cofactor imbalance between the conversion steps. XR normally has a higher affinity for NADPH than for NADH, whereas XDH uses only NAD+, which leads to an excessive accumulation of NADH and a shortage of NAD+ necessary for the XDH reaction, as shown in Fig. 1. Xylitol is formed to re-oxidize the NADH surplus resulting from those reactions, impairing ethanol yield. In this context, a plethora of strategies have been outlined to minimize xylitol formation and to improve cofactor availability, thus increasing ethanol yield. Protein engineering or mutagenesis techniques have been applied for that purpose, where coenzyme preference is altered—either of XR, by changing its preference from NADPH to NADH, or of XDH, from NAD + to NADP +. Also, metabolic engineering has proven to be an alternative to disturb cofactor availability in yeast to favor ethanol formation from xylose. For that, strategies usually rely on tuning the activity of endogenous cofactor-dependent enzymes, or the introduction of exogenous cofactor-producing ones, to favor the redox environment for the oxidoreductive xylose pathway in S. cerevisiae. However, it is relevant to note that xylitol is an important by-product in xylose metabolism, and its formation could be advantageous in some scenarios.

For instance, Jeppsson et al. (2006) expressed a mutant XR (K270M) from S. stipitis (Kostrzynska, Sopher & Lee, 1998) with increased affinity for NADH in a recombinant S. cerevisiae harboring S. stipitis XDH and overexpression of endogenous XKS1: higher ethanol yield and reduced xylitol formation were obtained. Other mutant S. stipitis XR (K270R (Watanabe et al., 2007a), K270G (Watanabe et al., 2007a), R276H (Watanabe et al., 2007c), N272D (Watanabe et al., 2007a), K270R/N272D (Watanabe et al., 2007c), N272D/P275Q (Runquist, Hahn-Hägerdal & Bettiga, 2010), and K270R/R276H (Watanabe et al., 2007c) with NADH preference were obtained and expressed in S. cerevisiae, showing the same results on ethanol and xylitol yields. Mutant XR has also been obtained from Candida tenuis (Kavanagh et al., 2002; Kavanagh et al., 2003; Petschacher et al., 2005; Leitgeb et al., 2005; Petschacher & Nidetzky, 2005), which could be used to balance the redox environment in xylose consuming S. cerevisiae. Petschacher & Nidetzky (2008) expressed a double mutant XR (K274R-N276D CtXR) in a recombinant S. cerevisiae and the resulting strain showed an increase in NADH utilization, which improved ethanol production and decreased xylitol secretion.

Endeavors in changing the specificity of the coenzyme of XDH, from NAD+ to NADP+, have also been described. Watanabe, Kodaki & Makino (2005) obtained the quadruple mutant ARSdR (D207A/I208R/F209S/N211R) that showed 4500-fold higher catalytic efficiency (kcat/Km) with NADP+ than wild-type S. stipitis XDH. The ARSdR mutant was expressed in recombinant S. cerevisiae strains under the control of a strong constitutive promoter (PGK1), together with S. stipitis XR, achieving increased ethanol yield (41%) and lower xylitol production (86%) (Watanabe et al., 2007b).

Because the shortage of NADPH results in less xylitol formation, reducing flux through the oxidative PPP—where this cofactor is normally generated, and wasteful CO2 is produced—is another approach for improved ethanol yield from xylose. The deletion of the ZWF1 gene, which encodes G6PDH and is responsible for the regeneration of this cofactor, and the deletion of GND1—one of the isogenes of 6-phosphogluconate dehydrogenase—were evaluated in strains containing the XYL1/XYL2 genes. Deletion of the GND1 gene resulted in an improvement in ethanol yield by 24% and a decrease in xylitol production. A ZWF1 null genotype, however, showed a significant increase in ethanol yield and a reduction in xylitol production. Although blocking the NADPH-producing PPP lowered xylitol formation, xylose fermentation was also reduced because XR reaction was mediated only by NADH (Jeppsson et al., 2002). To overcome this issue, overexpression of the fungal GDP1 gene encoding an NADP+ dependent GAPDH (glyceraldehyde-3-phosphate dehydrogenase)—not linked to CO2 production—along with ZWF1 deletion resulted in an improvement of ethanol yield of approximately 50% (Verho et al., 2003).

Metabolic engineering of ammonium assimilation has also been suggested as an alternative procedure to modulate redox metabolism and favor xylose fermentation in S. cerevisiae. Ammonium, often used as a nitrogen source in industrial fermentations with S. cerevisiae, is converted to glutamate by reaction with 2-oxoglutarate, catalyzed by an endogenous NADPH-dependent glutamate dehydrogenase encoded by GDH1 (Moye et al., 1985). Therefore, deletion of GDH1 and overexpression of GDH2, a NADH-dependent glutamate dehydrogenase, is expected to shift ammonia assimilation from being NADPH to NADH dependent, alleviating NADPH shortage for XR. Bearing that in mind, Roca, Nielsen & Olsson (2003) performed such metabolic engineering in a strain expressing XYL1, XYL2 and overexpression of endogenous XKS1. The final strain presented an increased ethanol yield and a 44% reduction of xylitol excretion. The same group tested the overexpression of the GS-GOAT complex (GLT1 and GLN1, participating in ammonium assimilation using NADH as cofactor) in the gdh1 deleted strain, which also resulted in an increased ethanol yield. Later on, comparative metabolic flux analysis revealed that, in a mutant strain with deleted GDH1 and overexpression of GDH2, a shift in the specific xylose reductase activity towards the use of NADH as a cofactor could explain the improved ethanol yield due to its benefit to cofactor imbalance (Grotkjær et al., 2005).

Meanwhile, the expression of heterologous enzymes that prevent cofactor imbalance has also been tested. Through a genome-scale metabolic modeling approach, Bro et al. (2006) found that the heterologous expression of GAPN gene, encoding a Streptococcus mutants non-phosphorylating NADP+-dependent glyceraldehyde-3-phosphate dehydrogenase, reduced the formation of xylitol by 33%, while increasing the production of ethanol by 24%. While glyceraldehyde-3-phosphate is converted to 3-phosphoglycerate in wild-type S. cerevisiae in a two-step NAD+-dependent reaction, S. mutants GAPN allows the same reaction avoiding competition for the cofactor used by XDH. Overexpression of NOXE, encoding a water-forming NADH oxidase from Lactococcus lactis, in a XR/XDH S. cerevisiae led to decreased xylitol formation and increased ethanol production during xylose fermentation (Zhang, Liu & Ding, 2012). NOXE provides an extra route for the oxidization of NADH resulting from the XDH reaction, thus rebalancing the cofactor environment to favor xylitol reduction.

Transporters

In the production of any metabolite from a cell, the first step is substrate assimilation. The efficient incorporation of substrate molecules into yeast cells is suggested as a critical factor for obtaining efficient biofactories (Hara et al., 2017). In yeast, sugar entry is facilitated by a family of sugar porters known as the major facilitator superfamily (MFS), present in different species in all kingdoms of nature (Marger & Saier, 1993; Rédei, 2008; Quistgaard et al., 2016). This transport of sugars occurs mostly by facilitated diffusion, being a passive transport mechanism of substances across the cell membrane (Jeffries, 1983). In S. cerevisiae, at least 18 genes encoding hexose transporters (HXT1-17) and galactose permease (GAL2) are found endogenously, however only HXT1-7 and GAL2 show active expression, with HXT 8-HXT17 being inactive (not transcribed) or expressed at very low levels (Özcan & Johnston, 1999; Hamacher et al., 2002; Sedlak & Ho, 2004).

Although native hexose transporters also have the ability to import pentoses, xylose-specific transporters are not found in S. cerevisiae, in such a way that their transport occurs inefficiently due to the lower affinity of such a transport system for this sugar (Sedlak & Ho, 2004; Subtil & Boles, 2012). Moreover, transporters that perform xylose assimilation suffer a strong inhibition in the presence of other sugars, especially glucose, and this repression is considered a limiting factor in mixed sugar fermentation, as most recombinant S. cerevisiae yeasts are unable to initiate xylose assimilation before glucose depletion (Bertilsson, Andersson & Lidén, 2008; Subtil & Boles, 2012; Farwick et al., 2014), causing a negative impact on the fermentation time of lignocellulosic biomass. Therefore, many studies have sought to design xylose-specific and/or glucose-insensitive transporters in order to obtain more efficient S. cerevisiae platforms for mixed sugar fermentation. In this context, cell platforms for sugar transporter characterization are obtained by deletion of native hexose transporters (hxt null), avoiding the interference of their effect on sugar transport analyses (Wieczorke et al., 1999; Boles & Oreb, 2018; Wijsman et al., 2019). hxt null strains do not exhibit the ability to grow on glucose as the sole carbon source, and have been used to characterize various endogenous hexose transporters, as well as those from different origins (Wieczorke et al., 1999; Hamacher et al., 2002; Young et al., 2011; Hara et al., 2017; Boles & Oreb, 2018). Information on the heterologous expression of transporters for xylose uptake and modification of endogenous transporters to improve affinity for this sugar in S. cerevisiae are summarized in Table 4.

Table 4 Sugar transport modifications for improved xylose consumption in S. cerevisiae using heterologous expression and endogenous modification strategies.

Strategy	Transporter genes	Mutation	Relevant genetic
background	Reported phenotype
improvement	Reference	
Heterologous expression	GXF1
(Candida intermedia)	–	MT8-1 - XKΔXI	Enhanced xylose consumption and ethanol production	(Tanino et al., 2012)	
–	TMB 3043 - Scheffersomyces stipitis XYL1 and XYL2 / XKS1 / RKI1 / RPE1 / TKL1 / TKL2 / TAL1 / gre3Δ	Under anaerobic conditions, increased xylose uptake and ethanol formation at low xylose concentrations	(Runquist et al., 2009)	
-	Scheffersomyces stipitis XYL1 and XYL2 / XKS1	2 times higher affinity for xylose	(Fonseca et al., 2011)	
GXS1
(Candida intermedia)	Phe3 8 Ile3 9 Met40	EX.12 - Scheffersomyces stipitis XYL1, XYL2 and XYL3 / hxt1-17Δ / gal2Δ	Growth on xylose but does not assimilate glucose	(Young et al., 2014)	
GXS1
(Candida intermedia)	F40	EX.12 - Scheffersomyces stipitis XYL1, XYL2 and XYL3 / hxt1-17Δ / gal2Δ	Increased affinity for xylose	(Young et al., 2012)	
XUT3
( Scheffersomyces stipitis )	E538K	EX.12 - Scheffersomyces stipitis XYL1, XYL2 and XYL3 / hxt1-17Δ / gal2Δ	Increased xylose uptake	(Young et al., 2012)	
SUT1
(Scheffersomyces stipitis)	-	Scheffersomyces stipitis XYL1 and XYL2 / XKS1	Increased xylose absorption capacity and ethanol productivity in fermentation	(Katahira et al., 2008)	
XUT1
(Scheffersomyces stipitis)	–	EBY.VW4000 - Scheffersomyces stipitis XYL1 and XYL2 / hxt1-17Δ / gal2Δ	Greater preference for xylose over glucose	(Young et al., 2011)	
CS4130
(Candida sojae)	–	EBY.VW4000 - Scheffersomyces stipitis XYL1 and XYL2 / XKS1 / hxt1-17Δ / gal2Δ	Xylose absorption at high substrate concentrations	(Bueno et al., 2020)	
MGT05196P
( Meyerozyma guilliermondii)	N360F	EBY.VW4000 - Scheffersomyces stipitis XYL1 and XYL2 / XKS1 / hxt1-17Δ / gal2Δ	Xylose transport without inhibition by glucose	(Wang et al., 2015)	
XITR1P
( Trichoderma reesei)	N326F	EBY.VW4000 - hxt1-17Δ / gal2Δ	High xylose transport activity / low growth in glucose	(Jiang et al., 2020)	
HXTB
(Aspergillus nidulans)	–	EBY.VW4000 - Scheffersomyces stipitis XYL1 and XYL2 / XKS1 / hxt1-17Δ / gal2Δ	Higher xylose growth and ethanol production	(Dos Reis et al., 2016)	
XTRD
(Aspergillus nidulans)	–	EBY.VW4000 - Scheffersomyces stipitis XYL1 and XYL2 / XKS1 / hxt1-17Δ / gal2Δ	Higher affinity for xylose	(Colabardini et al., 2014)	
AT5G17010 ( Arabidopsisthaliana)	–	BY4727 - Scheffersomyces stipitis XYL1 and XYL2 / XKS1	25% and 40% increase in xylose consumption	(Hector et al., 2008)	
AT5G59250 ( Arabidopsisthaliana)	–	
AT5G59250 ( Arabidopsis thaliana)	–	Scheffersomyces stipitisXYL1 (K270R) and XYL2 / XKS1 / TAl1 / TKL1 / RPK1 / RPE1 / gre3Δ	Did not present significant results in the kinetics of xylose absorption	(Runquist, Hahn-Hägerdal & Rådström, 2010)	
Endogenous modification	HXT7	F79S	BY4742 - Piromyces sp. XYLA / XKS1 / gre3Δ	Improved xylose absorption rates	(Apel et al., 2016)	
HXT7	N370S	AFY10X - Clostridium phytofermentans XYLA / TAL1 / TKL1 / RPI1 / RKI1 / XKS1 / hxk1Δ/ hxk2Δ / glk1Δ/ gal1Δ	Decreased inhibition by the presence of glucose	(Farwick et al., 2014)	
GAL2	N376F	AFY10X –Clostridium phytofermentans XYLA / TAL1 / TKL1 / RPI1 / RKI1 / XKS1 / hxk1Δ/ hxk2Δ / glk1Δ/ gal1Δ	Lost the ability to transport hexoses and has a greater affinity for xylose	
N376Y / M435I	SRY027 - XYLA / TAL1 / TKL1 / RPE1 / RKI1 / XKS1 / HXT2 / HXT7 / HXT9 / GAL2	Reduction of xylose consumption time by approximately 40%	(Rojas et al., 2021)	
T386A	DS69473 - Clostridium phytofermentans XYLA / TAL1 / TKL1 / RPE1 / RKI1 / XKS1 / gre3Δ/ hxt1-7Δ / gal2Δ / hxk1Δ, hxk2Δ, glk1Δ, gal1Δ	Increased xylose transport and reduced affinity for glucose	(Reznicek et al., 2015)	
HXT36 (Chimeric)	N367A	DS71054 - XYLA / TAL1 / TKL1 / RPE1 / RKI1 / XKS1 / gre3Δ / hxt1-7Δ / gal2Δ / glk1Δ/ hxk1Δ/ hxk2Δ/ gal1Δ	Xylose transport at high rates / efficient co-consumption of glucose and xylose	(Nijland et al., 2014)	
HXT11	N366	DS68625 - XYLA / TAL1 / TKL1 / RPE1 / RKI1 / XKS1 / hxt1-7Δ / gre3Δ / gal2Δ	Increased affinity for xylose compared to glucose / high transport rates / efficient co-fermentation of xylose and glucose	(Shin et al., 2015)	
HXT2	C505P	DS68625 - XYLA / TAL1 / TKL1 / RPE1 / RKI1 / XKS1 / hxt1-7Δ / gre3Δ / gal2Δ	Increased affinity and xylose transport flux at low concentrations of this substrate	(Nijland et al., 2018)	

One compelling approach to optimizing xylose uptake in S. cerevisiae is the insertion of heterologous specific xylose transporters from bacteria, fungi, yeasts or plants (Nijland & Driessen, 2020). In this context, many studies have focused on identifying those proteins in different species, especially from other xylose-fermenting yeasts such as Candida intermedia, S. stipitis and Meyerozyma guilliermondii. However, although expression of such transporters allowed growth on xylose in S. cerevisiae, glucose inhibition was still observed (Leandro, Gonçalves & Spencer-Martins, 2006; Runquist et al., 2009; Tanino et al., 2012). In parallel to heterologous expression, mutagenesis in native sugar transporters allowed enhanced xylose transport kinetics in the presence of glucose, as well as the co-utilization of both sugars (Li, Schmitz & Alper, 2016).

The high-capacity, low-affinity glucose/xylose facilitated diffusion transporter (GXF1), obtained from C. intermedia, showed a threefold improvement in transport kinetics and xylose utilization when expressed in S. cerevisiae; however, GXF1 improvements in xylose transport were only observed at low concentrations of this sugar. No changes in uptake rates at high concentrations of xylose were detected, suggesting that the expression of this specific transporter in S. cerevisiae would be beneficial only when the xylose concentration is not excessive (Runquist et al., 2009; Fonseca et al., 2011; Tanino et al., 2012). GXS1 is another sugar transporter identified in C. intermedia, where a F40 point mutation was located (Young et al., 2012), indicating that substitutions in F40 have a relationship with sugar transport dynamics and consequently can produce different phenotypes, including improved xylose transport.

Young et al. (2014) evaluated the sequence similarity of different heterologous transporters expressed in S. cerevisiae and reported a conserved amino acid motif (G-G/F-XXX-G) as responsible for monosaccharide selectivity in sugar transporters. An improved C. intermedia GXS1 was obtained by adding Phe38Ile39Met40 mutations, resulting in a pentose transporter with a slight increase in xylose uptake rate; nevertheless, transportation remained inhibited by glucose.

S. stipitis has also been widely used to prospect xylose transporters due to its natural ability to ferment this sugar. Many transporters from this species were analyzed and expressed in hxt null S. cerevisiae mutants, among them SUT1, SUT2 and SUT3 (Weierstall, Hollenberg & Boles, 1999); XUT1 and XUT3 (Young et al., 2011). The SUT1 transporter, when expressed in a strain of S. cerevisiae, showed improvement in xylose transport and ethanol productivity in fermentation (Katahira et al., 2008). The XUT3 transporter, on the other hand, had an average efficiency in transporting sugars, but with a greater preference for xylose (Young et al., 2011). Young et al. (2012) suggested that the E538K mutation in XUT3 is responsible for improved xylose affinity, in addition to improved growth at low xylose rates.

Other fungi have also been a source of efficient xylose transporters when expressed in recombinant S. cerevisiae. Bueno et al. (2020) used an evolutionary approach combined with analysis of diverse microbiomes to identify new xylose transporter candidates. In the study, the CS4130 transporter from Candida sojae was identified and showed functional expression in S. cerevisiae at high xylose concentrations, revealing an appealing alternative for industrial fermentation of that pentose. The MGT05196P transporter identified in M. guilliermondii also showed elevated xylose transport activity in S. cerevisiae, and mutant N360F was able to transport xylose without any glucose inhibition (Wang et al., 2015). From the xylose-consuming filamentous fungus Trichoderma reesei, the XITR1P was reported as a xylose transporter with better efficiency than the endogenous S. cerevisiae transporter GAL2. Through site-directed mutagenesis it was indicated that the N326F amino acid mutation is highly correlated to xylose-uptake activity, and its expression in S. cerevisiae conferred high efficiency in transporting this sugar, while being insensitive to glucose (Jiang et al., 2020). Many other transporters have been identified in different origins: HXTB and XTRD (Aspergillus nidulans) are two such examples (Colabardini et al., 2014; Dos Reis et al., 2016).

In Arabidopsis thaliana, genes encoding sugar transporters AT5G17010 and At5g59250 were expressed in recombinant S. cerevisiae containing the genetic modifications for xylose consumption, and the consumption of this pentose was analyzed in fermentations. Strains expressing the AT5G17010 and AT5G59250 transporters consumed 25% and 40% more xylose, respectively, than the control strain (Hector et al., 2008). However, in another study using different concentrations of xylose, no significant values were obtained in the transport of the xylose transporter AT5G59250 compared to the control strain (Runquist, Hahn-Hägerdal & Rådström, 2010).

Although many studies have focused on the expression of heterologous xylose transporters in S. cerevisiae, the low activity and stability of such exogenous proteins, as well as the fact that most of these transporters exhibit competitive inhibition by glucose, limits their use in fermentations with co-consumption of sugars (Hou et al., 2017). Thus, another widely used strategy is the expression of endogenous transporters modified to reconnect sugar affinity. Although recombinant strains exhibit the ability to ferment xylose as the sole carbon source, when mixed glucose and xylose fermentations are performed, xylose is consumed only after glucose depletion because the affinity of endogenous transporters for glucose is much higher than that of xylose, leading to slow metabolization of xylose in the presence of this hexose, even at low concentrations of this sugar (Subtil & Boles, 2012; Hou et al., 2017). Several studies have sought to improve the ability of simultaneous sugar metabolization in recombinant strains, requiring a reduction in the affinity of hexose transporters for glucose, as well as an increase in their affinity for xylose (Farwick et al., 2014).

In S. cerevisiae, endogenous HXT1-7 transporters along with GAL2, are responsible for the facilitated diffusion of xylose monosaccharides (Sedlak & Ho, 2004). Many studies have used different methodologies aiming to improve the ability of xylose/glucose co-metabolism by increasing the affinity of hexose transporters to xylose in modified strains. Among the strategies used for this purpose are random mutagenesis, genetic shuffling, evolutionary engineering, and overexpression, which have identified several mutant xylose transporters that do not undergo strong inhibition by glucose (Farwick et al., 2014; Young et al., 2014; Shin et al., 2015; Li, Schmitz & Alper, 2016).

In this context, using an ALE strategy, a platform for the evaluation of xylose transporters that lack inhibition by glucose was developed. Through this approach and error-prone PCR-based mutagenesis, two glucose-insensitive mutant xylose transporters, HXT7 (N370S) and GAL2 (N376F), have been identified (Farwick et al., 2014). In another study, an endogenous chimeric transporter (HXT36) was constructed using the endogenous transporters HXT3 and HXT6. After the evolutionary engineering of a strain expressing the synthetic HXT36 transporter, an N367A mutation was identified that generated increased affinity for xylose (Nijland et al., 2014). In another evolutionary engineering study, an F79S mutation in HXT7 resulted in improved D-xylose uptake (Apel et al., 2016). Shin et al. (2015) (Shin et al., 2015) identified a mutation on residue N366 in HXT11 in a recombinant S. cerevisiae with gene knockouts HXT1-7 and GAL2 that altered the specificity of the glucose transporter for xylose and enabled improved co-fermentation of these sugars. Another mutation identified was C505P which resulted in a 3-fold improvement in the xylose affinity of HXT2 (Nijland et al., 2018).

Although many mutations have been identified as contributors to the affinity change in hexose transporters, a conserved asparagine residue has been identified in several studies at positions 360, 366, 367, 370 and 376 in Meyerozyma guilliermondii MGT05196P (Wang et al., 2015), and endogenous HXT11 (Shin et al., 2015), HXT36 (Nijland et al., 2014), HXT7 and GAL2 (Farwick et al., 2014), respectively. This asparagine residue was mutated to different amino acids, causing a decreased affinity for glucose and, in some cases, an increased affinity for xylose, indicating this as an important target for mutagenesis. Later a, GAL2 N376Y/M435I double mutant was obtained, reported to be completely insensitive to competitive inhibition by glucose, and presented an improved ability to transport xylose upon expression in recombinant S. cerevisiae (Rojas et al., 2021). Another mutation identified in GAL2 was threonine at position 386 (T386A), allowing for increased xylose transport and reduced glucose sensitivity, as well as co-consumption at reduced substrate concentrations (Reznicek et al., 2015).

Overexpression of hexose transporters has also been shown as another compelling approach to improve xylose uptake. Different studies have proven that overexpression of the endogenous hexose transporters, HXT and GAL2, can also provide improvements in the rate of xylose uptake in recombinant S. cerevisiae (Tanino et al., 2012; Gonçalves et al., 2014).

Conclusions

Metabolic engineering has been used to optimize microorganisms through targeted alteration in simple cellular characteristics. The genetic alterations listed in this document could be rationally introduced in yeast cells for improved xylose metabolism. In S. cerevisiae, such interventions have contributed to increased growth rates and xylose assimilation, ultimately leading to better fermentation performance. However, the need to upgrade this phenotype foments other engineering approaches that could result in highly efficient strains. Evolutionary engineering, associated with chemical mutagenesis techniques, genome shuffling, genomic library screenings or transposon mutagenesis, are feasible approaches to develop mutant strains with enhanced xylose consumption and increased ethanol production rates. Other complex approaches—such as the omics: genomics, transcriptomics, metabolomics and fluxomics—directly contribute to advancing the understanding of different phenotypes at the molecular level through the identification of new genetic targets responsible for the enhancement of phenotypes.

However, despite the success in approaches used to obtain xylose assimilating S. cerevisiae, the understanding of the metabolism, regulation and signaling pathways involved in xylose consumption is still limited. There are hidden features of xylose metabolism that need to be identified to optimize fermentation processes. New approaches should be sought to identify non-obvious gene targets and to analyze the role of essential genes for the xylose consumption phenotype, as well as to evaluate the optimal expression level of genes directly and indirectly involved in xylose metabolism. Ultimately, advances in pentose metabolism in S. cerevisiae are expected to boost biotechnological routes for the full exploration of lignocellulosic biomass in a low-carbon economy.

We express our appreciation to Prof. Dr. Marcelo Falsarella Carazzolle for his valuable suggestions and constructive advice in helping us develop this paper. We also thank Francisco Martin Rivera for providing linguistic improvements.

Additional Information and Declarations

Competing Interests

Author Contributions

Data Availability

The authors declare there are no competing interests.

Beatriz de Oliveira Vargas performed the experiments, analyzed the data, prepared figures and/or tables, and approved the final draft.

Jade Ribeiro dos Santos performed the experiments, prepared figures and/or tables, and approved the final draft.

Gonçalo Amarante Guimarães Pereira conceived and designed the experiments, authored or reviewed drafts of the article, and approved the final draft.

Fellipe da Silveira Bezerra de Mello conceived and designed the experiments, authored or reviewed drafts of the article, and approved the final draft.

The following information was supplied regarding data availability:

This is a literature review.

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
