# Peer review of "An atlas of rational genetic engineering strategies for improved xylose metabolism in Saccharomyces cerevisiae"

_PeerJ, doi:10.7717/peerj.16340_

## Round 0.1 · original submission · Minor Revisions

Your review article has been assessed by two independent reviewers, who are largely positive about the paper but suggest ways in which it can be improved. I agree with reviewer #2 that it would be greatly enhanced by the use of a professional proof-reading service, or at least a thorough review by a fluent English speaker, as there are many grammatical errors, spelling mistakes and use of incorrect words throughout the manuscript that detract from its quality. I would also suggest the use of more informative subheadings; for example instead of "The isomerase" consider using something like "Strategies to clone and express isomerase in Saccharomyces". This would allow the reader to skim through the article and focus on areas of particular interest. It would also improve the general flow of information to have it organised in this way.

I also agree that more could be made of Figure 1 as a means of providing an anchor for the paper, where various aspects of metabolism can be referred to and described/discussed as one works through the article.

Please provide a point-by-point rebuttal of the reviewer comments with your revised manuscript.

·

Basic reporting

The review describes the genetic engineering strategies that can be used to improve xylose metabolism in S. cerevisiae. The review includes different pathways or strategies which could be employed, including the oxidoreductive pathways, isomerase, manipulation of endogenous enzymes, tuning cofactor binding, and sugar transporters, and rewiring of metabolic networks through deletion, overexpression and regulatory fine tuning. The details on the genes and pathways that can be manipulated to bring about effective improvements in xylose metabolism were very clearly described, was easy to understand for people with biochemistry and genetics knowledge and would be useful for anyone wanting to learn more about this field. Each section provides comprehensive quantitative information on the effectiveness of each strategy. These quantitative improvements were also summarised in a series of tables, which would serve as handy references for anyone wanting to utilise these data. I had checked a list of recent related review papers on PubMed, the closest I could find is the following one:

Moysés DN, Reis VC, de Almeida JR, de Moraes LM, Torres FA. Xylose Fermentation by Saccharomyces cerevisiae: Challenges and Prospects. Int J Mol Sci. 2016 Feb 25;17(3):207. doi: 10.3390/ijms17030207. PMID: 26927067; PMCID: PMC4813126.

This paper I’m reviewing now provides unique and detailed information and more recent developments the previously published paper by Moyés et al. 2016 above.

Experimental design

The search strategies and keywords used for selecting and compiling a list of papers to perform systematic review was comprehensive and included 160 papers that covers the field of xylose metabolism and genetic engineering in S. cerevisiae. The search strategies are easy to follow and replicable. The survey methodology is appropriate for this systematic literature review.

Sources were adequately cited and quoted or paraphrased as appropriate.

The review is organized logically into coherent paragraphs and subsections (e.g. oxidoredictive pathways, isomerase, transporters, deletion, overexpression etc…).

Validity of the findings

The review paper reads well, and the information presented on how to improve xylose metabolism were logical, easily interpretable and were well supported by literature citations. There is enough details and references that someone working in a yeast lab could easily follow similar protocols, manipulate a selection of pathways describe in the review paper and design an engineered yeast strain that metabolise xylose. There is a good coverage of strategies, from accelerated evolution, error-prone PCR, synthetic genetic array, genomics, transcriptomics etc.. and other strategies, there is a comprehensive list of the important yeast genetic manipulation techniques listed in the review.

From someone like me who does not direct working in the xylose metabolism field, I will suggest the author provide some brief comments on whether the quantitative yield data were obtained from small-scale laboratory studies or in large-scale industrial settings. Similarly, it could be useful to comment on how well these quantitative improvements will translate to industrial scale setting. For example, how often would there be glucose left over from first round of fermentation in industrial settings and whether we would need have to certain changes (e.g. inhibit glucose inhibition by transporters, or other necessary improvements), before the yeast can be used viably in industrial settings. I will leave this as a suggestion only and will be happy to have the paper published without addressing this comment.

The conclusion is clear and to the point and highlights the need to understand more about the intricacies of the xylose metabolism system in order to further improve ethanol yield.

Additional comments

I have some suggestions with regards to clarity of some sentences, and possible grammatical errors or typos. I will suggest using spellcheck function on the entire text and a careful read through of the text to check for typos. Since english is not my first language and I do make spelling and grammatical mistakes, so I will like to offer these as suggestions to be further investigated the authors.

Line 444 to line 449. The sentence is a bit long and I feel that it is not very clear or easy to understand. I suggest simplifying the sentence or breaking up the sentence.

Line 55: I'm not sure if it is more appropriate to use the word oxidoreductive or oxireductive. I could be wrong. Oxireductive is used throughout the text, I would suggest double checking please?

Line 81: Change 'Survay' to 'Survey'

Line 610: etanol to ethanol

Line 699: mutagenisis to mutagenesis

Line 740: incentive to insensitive

Reviewer 2 ·

Basic reporting

Overall, the review manuscripts covers sufficient study revolving the proposed title and scope. However, the current draft needs to be improved and some comments need to be properly addressed before it can be accepted to be published in PEERJ.

1.The review does cover most aspect and across different strategies
2. To the best of my knowledge, limited review such as this in recent years have been done
3. Overall, the introduction is sufficient, holistic and would benefit the readers, however there are areas that needed to be improved esp linking and emphasizing more of the figure 1

Experimental design

Sufficient and Yes. However, please refer to detail comments to improve and amends some of the points being raised.

Validity of the findings

Please refer to the detail comments.

Additional comments

PeerJ Manuscript Review
An atlas of rational genetic engineering strategies for improved xylose metabolism in Saccharomyces cerevisiae (#85439)

Overall, I find the review quite interesting and would benefit a lot researchers and related readers. However, there are few areas needs to be improved and justified. Some of the review of sections are extensive and details but most time is just brief and simple reporting. Some sections/paragraph were sound superficial with no real justification in in-depth analysis from the authors. No real deductions on certain relationship between cause and effects. Format, English writing, grammar, typo error and references need to be checked. I would suggest that this to be sent to professional proof-reader that can double check on this.


Introduction

• L 46 – “hability” – is this even a correct and formal word? Seen this few times

• L 81 – typo

Main body

• L82 – all acronym should be spelt in full first – NCBI
• L 96 – change the term oxireductive (not common/unusual) to oxidoreductive/oxido-reductive
• L121 – Many studies, but yet you pun only 2 references. You stated ME is being applied, but no example is being shown and was not explain properly. Only briefly
• L210-211 – Please double check the correct form of writing gene names for microorganism. Common style would be small italicised letters. Capital italicised letter would be normally reserve for human genes.
• L219-220 – need to be more specific in the increase for review article. Decrease/Increase by how much (value or percentage)
• L 245 – if you stated it affect the growth, what was the reason that could affect its growth. It’s important to review its cause and effects on its metabolic change once you modify >1 genetic changes
• L 277 – “yield” (one example of many) – there are many typo errors and grammatical mistakes in the manuscript. I would highly suggest sending the manuscript to proof-reader to eliminate these mistakes.
• L 293 – reference yeast refers to what? What about the improvement on MNI1 and RPA49?
• L 295 – if it’s under OE, why mention these genes in this section. Does it change the context of your point if you were to delete this? I don’t see much contribution put it here and might just added redundancy and confusion.
• L306 – 305 – How so the deletion improved and why? How this non-essential genes contribute. You need to discuss/justify rather than just reporting it.
• L 385 – maintain your citation style (Suggest change to, Shin et al., (2021). Please change throughout.
• L418 – 435 – The impression of OE on XK seems contradictory on the host. It is not clear the on the take that the authors would agree or emphasize on the effects of XK on xylose metabolism and ethanol production or detrimental to the host. Suggest refining the whole paragraph with more decisive and clear intentions.
• L530-534 – might need to provide brief explanation how weak promoters such as GND2 and HXK2 (from which strain) can cause OE. If its at any level or OE, why this is under fine tuning and not under OE section? How controlling the level of OE (could you provide in number how many fold since this is fine tuning expression) can affect xylose fermentation and ethanol production?
• L 566 – resistance and stability against what? What type of stressful condition being applied? How this affect to the xylose fermentation and ethanol?
• L 577 – 585 – The relationship better promoter and terminator needs to be explained in more detail. Why only CPS1 better and not the other 3. Previously it was mentioned that terminator involved in “I) transcriptional termination and II) post-transcriptional regulation” . How this processed affect the enzyme/protein structure or its activity that would relate to increase in xylose fermentation.
• L593 – last reaction? Refer to what?
• L621 – 622 – Inconsistent writing on value/percentage increase and decrease. Some stated and some aren’t. Highly recommend that this should be consistent where the value should be stated throughout the manuscript to provide impression how much changes coming the gene modification.


Acknowledgement

• As far as my understanding, this should be strictly for grants, facilities, programmes or other non-living contributions. If Prof. Dr. Marcelo Falsarella Carazzolle, contribute that much, she should be one of the authors.
References

• Should be in alphabetical order, unless PEERJ has its own format that I am not aware.
• Need to check the format thoroughly. Strain names on written scientifically. If the authors use software, please do not completely rely on it not to make mistakes.
• Maintain consistent style of references. Capital each letter? Only capital first starting title letter?
• Some use et al. and some would list all the names.
• Highly recommend this to be sent to reputable proof-reader to ensure format and writing error-free manuscript

Figure

Figure 1 – Figure should be clear and descriptive. The authors should add brief description in the caption. This figure is such a strong statement in this review, yet it is briefly described in the manuscript itself (L 56). Please add strong description regarding the figure and how this strategy illustrate in figure relate to the body part of you review manuscript. This should link up with L 255 – 259 for better understanding to the reader on the way you approach organizing the flow of the review.

Annotated reviews are not available for download in order to protect the identity of reviewers who chose to remain anonymous.

---

## Round 0.2 · Minor Revisions

Thank you for revising your manuscript, which is now greatly improved. As noted by Reviewer #2 some issues with English expression remain, and while these are generally minor it would be beneficial to have these fixed in order to improve readability. Please therefore have the manuscript reviewed by a proofreading service or fluent English speaker prior to resubmission.

**Language Note:** The Academic Editor has identified that the English language must be improved. PeerJ can provide language editing services - please contact us at [email protected] for pricing (be sure to provide your manuscript number and title). Alternatively, you should make your own arrangements to improve the language quality and provide details in your response letter. – PeerJ Staff

·

Basic reporting

I have reviewed the revised manuscript and thinks that the revised manuscript is satisfactory for all the basic reporting criteria, as per my previous comments. This is with the small exception that there seems to be some errors in the way citations are formatted in the new version. For example in line 197, the cited authors were repeated in the bracket "Parachin et al. (2011) (Parachin & Gorwa-Grauslund, 2011),". I will suggest the authors tidy these up prior to publication, perhaps as part of the final editing processes. I had read all the comments by both reviewers, reviewed the author's responses and the track changes in the revised manuscript. I thank the authors for addressing all of my comments appropriately. I do not have further comments other than the formatting problem with citations mentioned above.

Experimental design

I have reviewed the revised manuscript and thinks that the revised manuscript is satisfactory for all the study design criteria, as per my previous comments.

Validity of the findings

I have reviewed the revised manuscript and thinks that the revised manuscript is satisfactory for all the criteria in the "validity of the findings" section, as per my previous comments. My question about industrial studies has been appropriately addressed by the authors in the text (Lines 81-84).

Additional comments

None.

Reviewer 2 ·

Basic reporting

I believe the authors have made the necessary changes to improve the current manuscripts and have answered all the comments.

The authors claimed that they have sent this to proof-reader. I must insist that some of the flow from the manuscripts in very much the same. It would be best if the authors would provide proof-reader invoice or certificate. If there is none, I would highly recommend to send to PEERJ professional proof-reader team to address this issue.

Experimental design

no comment

Validity of the findings

no comment

Additional comments

no comment

---

## Round 0.3 · accepted · Accept

Thank you for your resubmission and for improving the English in your manuscript. I have reviewed it and am satisfied with the changes and feel this is now ready for publication.